# Unravelling the contribution of potential evaporation formulation to uncertainty under climate change

Thibault Lemaitre-Basset[1, 2], Ludovic Oudin[1], Guillaume Thirel[2], and Lila Collet[2, 3]

[1]Sorbonne Université, CNRS, EPHE, UMR 7619 METIS, Case 105, 4 place Jussieu, F-75005 Paris, France
[2]Université Paris-Saclay, INRAE, HYCAR research unit, Hydrology Research Group, Antony, France
[3]Now at EDF R&D, OSIRIS Department, 7 boulevard Gaspard Monge, 91120 Palaiseau, France

**Correspondence:** Thibault Lemaitre-Basset (thibault.lemaitre-basset@sorbonne-universite.fr)

**Abstract.** The increasing air temperature in a changing climate will impact actual evaporation and have consequences for water resources management in energy-limited regions. In many hydrological models, evaporation is assessed by a preliminary computation of potential evaporation (PE) representing the evaporative demand of the atmosphere. Therefore, in impact studies the quantification of uncertainties related to PE estimation, which can arise from different sources, is crucial. Indeed, a myriad of PE formulations exist and the uncertainties related to climate variables cascade into PE computation. So far, no consensus has emerged on the main source of uncertainty in the PE modelling chain for hydrological studies. In this study, we address this issue by setting up a multi-model and multi-scenario approach. We used seven different PE formulations and a set of 30 climate projections to calculate changes in PE. To estimate the uncertainties related to each step of the PE calculation process (namely Representative Concentration Pathways, General Circulation Models, Regional Climate Models and PE formulations), an analysis of variance decomposition (ANOVA) was used. Results show that mean annual PE would increase across France by the end of the century, from +40 to +130 $mm.y^{-1}$. In ascending order, uncertainty contributions by the end of the century are explained by: PE formulations (below 10%), then RCPs (above 20%), RCMs (30-40%) and GCMs (30-40%). However, under a single scenario, the contribution of the EP formulation is much higher and can reach up to 50% of the total variance. All PE formulations show similar future trends since climatic variables are co-dependent to temperature. While no PE formulation stands out from the others, in hydrological impact studies the Penman-Monteith formulation may be preferred as it is representative of the PE formulations ensemble mean and allows accounting for climate and environmental drivers co-evolution.

## 1 Introduction

Ongoing climate change results in regional changes of precipitation regimes and a global increase of air temperature (Intergovernmental Panel on Climate Change, 2014a). As a consequence, the increase in evaporation has been highlighted as a potential key risk that may decrease streamflow and water resources, particularly in Europe and arid environments (Intergovernmental Panel on Climate Change, 2014b). However, the relationship between air temperature and evaporation increase is not straightforward. This relationship is highly dependent on water availability and atmospheric feedbacks (Boé and Terray, 2008). Indeed,

some studies pointed out a decreasing trend for evaporation in observed records, despite an increasing trend in air temperature,
due to soil moisture limitation (Jung et al., 2010) or atmospheric feedbacks such as increasing air moisture (Allen et al., 1998).

In many crop water requirements and hydrological models, evaporation is assessed by a preliminary computation of potential evaporation (PE) representing the evaporative demand of the atmosphere and used as inputs in the models. A large panel of formulations exist for PE estimation, from empirical temperature-based methods to more integrative ones, based on energy budget. In impact studies, the choice of an empirical temperature-based method is often motivated by the limited confidence in climate
variables other than air temperature (Wilby and Dessai, 2010; Dallaire et al., 2021), while the choice for more physically-based formulations is guided by the will to explicitly take into account interactions between radiative and aerodynamic variables that may co-evolve under climate change (McKenney and Rosenberg, 1993; Donohue et al., 2010). Frameworks to assess PE formulation validity in climate change impact studies concentrated on the ability to reproduce past and sometimes extreme events (Prudhomme and Williamson, 2013), but assuming that models may represent past and future climates equally well is
difficult to verify. Here, we propose to assess the contribution of PE formulations to the overall uncertainty of projections by testing several formulations under several climate projections. Since PE formulations are not the only source of uncertainty in impact modelling chains and since PE uncertainties also stem from the uncertainty conveyed by General Circulation Models (GCMs), Regional Climate Models (RCMs) and/or downscaling methods, quantifying the contribution of PE formulations to the total uncertainty of PE estimates is deemed crucial.

Previous studies that examined the contribution of PE formulations to the total uncertainty of PE projections show rather divergent results. Hosseinzadehtalaei et al. (2016) considered seven alternative PE formulations to the Penman-Monteith under a large set of 44 GCMs-RCPs couples over Belgium. They found that RCPs, GCMs and PE formulations show balanced contributions to the total PE uncertainty. McAfee (2013) compared three different PE formulations in the North American Great Plains and showed that the Hamon formulation presents higher increase of PE than Penman and Priestley-Taylor formu-
lations. Wang et al. (2015) showed that the Penman-Monteith formulation leads to a higher increase of PE than the Hargreaves formulation in China.

Regarding hydrological projections, Bae et al. (2011), Seiller and Anctil (2016), Williamson et al. (2016) and Milly and Dunne (2016, 2017) showed how future streamflow anomalies can be dependent on the choice of PE formulation. Conversely, Koedyk and Kingston (2016) and Thompson et al. (2014) found that PE formulation adds a minor contribution to the total
uncertainty of streamflow anomalies in the Mekong River and over New Zealand. Kay and Davies (2008) found that climate models bring the most uncertainties but pointed out that hydrological impacts can be quite different depending on the PE formulation used across Great Britain. Vidal et al. (2013) studied the differences between two PE formulations over the French Alps, and found a very significant contribution of the PE formulations to the total uncertainty of the projections.

These rather mixed results may originate from several choices made by the authors. First, the studies did not use the same
ensemble of PE formulations. For example, Kay and Davies (2008) used two PE formulations while Koedyk and Kingston (2016) and Milly and Dunne (2017) used up to eight different formulations including both empirical and physically-based formulations derived from an energy balance. Second, not all studies used the same radiative forcing scenarios and were often limited to a single scenario. For example, Koedyk and Kingston (2016) limited their study to a warming of 2°C. However, other

studies have focused on the most pessimistic greenhouse gas emission scenario associated with the greatest global warming.

For instance, Milly and Dunne (2017) only explored the CMIP5 RCP 8.5 scenario, and before them Kay and Davies (2008) only used SRES A2. Third, results appear to be linked to the study spatial scale, depending on whether the authors are interested in climate impact at the global scale (e.g. Milly and Dunne, 2016, 2017), the regional scale (e.g. Kay and Davies, 2008), or the local scale (e.g. Koedyk and Kingston, 2016). Vidal et al. (2013) only focused on mountainous areas, where all PE formulations are questionable, because all physical processes involved under cold and mountainous regions are not represented (part of the

snow cover disappears by sublimation without melting). Finally, studies differ on the variable of interest on which the total uncertainty is computed: it can be either the PE estimate itself or streamflow simulated by a hydrological model. Assessing the PE uncertainty on the resulting streamflow simulations is legitimate for case studies where water resources are assessed but it largely complicates the analysis since the sensitivity of streamflow simulations to PE inputs is conditioned both by hydrological model parameterization and by the climatic settings of the studied area (Koedyk and Kingston, 2016).

In this study, we use a comprehensive framework, including a large variety of seven PE formulations under several scenarios and using a large set of thirty CMIP5 GCM/RCM outputs. The sensitivity of impact models to PE is not addressed in this study, we focus our analysis on PE and assess the contribution of the formulations to the total PE uncertainty over a large domain, France. First, the analysis will focus on the future change in potential evaporation under climate change. Second, the total uncertainty on projected PE will be partitioned and quantified among all uncertainty sources (RCPs, GCMs, RCMs, and

PE formulations), then uncertainty for RCP 8.5 only.

## 2   Material and method

### 2.1   Climate projections

Three different RCPs (RCP 2.6, RCP 4.5 and RCP 8.5) were used to account for the uncertainty in future greenhouse gas emission trajectories and climate variables from thirty GCM/RCM couples from EURO-CORDEX (Jacob et al., 2014) (Table

1). The use of several models at each step allowed for a more robust quantification of the uncertainties stemming from each step. The modelling chain describes a pathway from different RCPs to an impact model (here PE formulations), with a succession of models, whose simulation outputs feed the next model. The figure 1 represents the modelling chain used for this study with each modelling step, namely RCPs, GCMs, RCMs and PE formulations. The reference period used with climate projections to compute PE anomalies is 1976-2005 and projections were analysed over 1976-2099. In practice 30-year periods are used

for climate impact studies, and the EURO-CORDEX simulations using the climate change scenarios cover the 2006-2100 time period. For all RCPs/GCMs/RCMs combinations only a realization is available. It hinds quantification of the internal variability as uncertainty sources. However, as anomalies are computed over long time slices, we can assume that natural climate variability has a limited impact. All data were available at the daily time step.

Table 1. The available climate projection data. The numbers (2.6, 4.5 and 8.5) refer to the RCPs used by the GCM (rows)/RCM (columns) pairs. Empty boxes or missing RCPs show the absence of data.

| GCM/RCM | Aladin63 | Racmo22E | WRF381P | RCA4 | RegCM4 | CCLM4-8-17 | REMO2009 | HIRHAM5 | REMO2015 |
|---|---|---|---|---|---|---|---|---|---|
| IPSL-CM5A | | | 4.5 8.5 | 4.5 8.5 | | | | | |
| CNRM-CM5 | 2.6 4.5 8.5 | 2.6 4.5 8.5 | | | | | | | |
| EC-EARTH | | 2.6 4.5 8.5 | | 2.6 4.5 8.5 | | | | | |
| HadGEM2-ES | | | | | 2.6 8.5 | 4.5 8.5 | | | |
| MPI-ESM-LR | | | | | | 2.6 4.5 8.5 | 2.6 4.5 8.5 | | |
| NorESM1-M | | | | | | | | 4.5 8.5 | 2.6 8.5 |

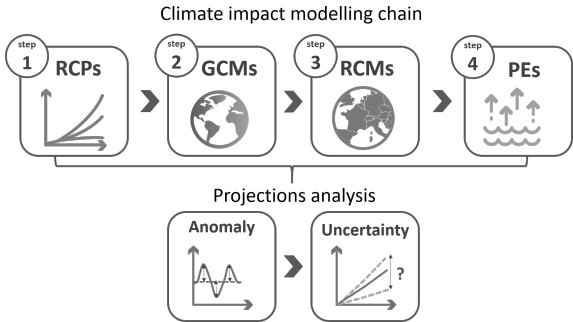

Figure 1. Diagram representing the modelling chain for the study of the impact of climate change on potential evaporation.

## 2.2 PE formulations

Seven PE formulations were selected in this study (see Table 2). This selection was made to represent diverse ways of estimating PE, including physically-based methods derived from the energy balance and empirical methods. In this study, all formulations were applied at a daily time step.

The Penman and Penman-Monteith formulations are often referred to as combinational methods since they are derived from the energy budget coupled with aerodynamic considerations. While the Penman formulation is recommended for open water evaporation estimation, the Penman-Monteith formulation was proposed to estimate the potential evaporation from a reference crop. These two formulations are widely used in crop water requirements and hydrological models since they make full use of currently measurable climate variables, under a physically-derived framework. For climate change impact studies, these formulations allow to take into account possible interactions and feedbacks between climate variables. The Priestley-Taylor formulation is a simplification of the Penman equation that allows estimation of PE with only radiative climate variables. This formulation does not make use of aerodynamic climate variables, which are highly uncertain in climate change impact studies. The Morton formulation was recommended by McMahon et al. (2013) for hydrological modelling. Based on the Priestley-

**Table 2.** The seven formulations used to compute PE. The column 'Climate variables' refers to the input data required in each PE formulation. $R_n$ is net radiation, $rh$ is relative humidity, $u_2$ is 2-m wind speed and $T_a$ is 2-m air temperature. Full equations are available in the provided R code (see code availability section). Penman-Monteith method from FAO 56; Hamon formulation use theorical sunshine hours (Oudin et al., 2005; Almorox et al., 2015). RRM = Rainfall-Runoff Modelling.

| Name and notation | Sources | Type of application | Climate variables |
|---|---|---|---|
| Penman | Penman (1948); Allen et al. (1998) | RRM / open water surface | $R_n, T_a, u_2, rh$ |
| Penman-Monteith | Monteith (1965); Allen et al. (1998) | RRM / crop evapotranspiration | $R_n, T_a, u_2, rh$ |
| Priestley-Taylor | Priestley and Taylor (1972) | RRM | $R_n, T_a$ |
| Morton | Morton (1983); McMahon et al. (2013) | RRM / catchment water balance | $R_n, T_a, rh$ |
| Oudin | Oudin et al. (2005) | RRM | $T_a$ |
| Hamon | Hamon (1963); Oudin et al. (2005) | RRM | $T_a$ |
| Hargreaves | Hargreaves and Samani (1985); Allen et al. (1998) | RRM / crop evapotranspiration | $T_a$ |

Taylor formulation, it includes an iterative estimation of a so-called equilibrium temperature, which better represents the surface temperature. Oudin, Hamon and Hargreaves formulations use air temperature only. Proxies of solar radiation are implicitly included in these formulations either through extraterrestrial radiation estimation or using empirically-derived equations that relate radiation to mean or amplitude of daily air temperature. For climate change impact studies, these formulations are interesting since air temperature is probably the least uncertain variable in climate projections. However, the absence of other climate variables is questionable under climate change since feedbacks between climate variables exist, e.g. the dimming effect making radiation decrease while air temperature increases. Shaw and Riha (2011) pointed out higher future PE amounts with the temperature-based formulations, compared to other formulations, over five predominantly hardwood forest sites at differing latitudes in the eastern United States.

### 2.3 Quantifying and partitioning projected PE uncertainties

A Bayesian data augmentation technique, the QUALYPSO method (Evin et al., 2019; Evin, 2020), was applied to deal with the lack of balance in terms of representation within the combinations of climate models (GCMs/RCMs) and RCPs (see gaps in Table 1). The data set available for this study is composed of 30 RCP/GCM/RCM chains, multiplied by seven PE formulations. The Bayesian data augmentation process of the QUALYPSO framework fills a complete and balanced matrix of climate projections composed of 162 members (3 RCPs*6 GCMs*9 RCMs), multiplied by seven PE formulations. This process results in a balanced data set, which is essential to correctly assess the contribution of each modelling step and to avoid inducing a biased estimation of the variance explained by a specific modelling step over or under represented. This framework was successfully applied by Lemaitre-Basset et al. (2021) to analyse projected hydrological uncertainties with an incomplete ensemble of projections. ANOVA methods are frequently used to quantify the contribution of different models to

total uncertainty. They rely on respective variance contributions of the different modelling chain steps to the total variance. The analysis of variance can be performed with a time series approach as mentioned by Hingray and Saïd (2014), which considers the quasi-ergodicity of climate variables for the long term. This approach was also used by Lafaysse et al. (2014) and Vidal et al. (2016) who added the downscaling and hydrological modelling steps to the modelling chain in the evaluation of uncertainties.

In the present study, to quantify and partition the total uncertainty on the projected changes among the different modelling steps, the QE-ANOVA framework was chosen (Hingray and Saïd, 2014; Hingray et al., 2019). The QE-ANOVA method allows decomposing the total variance of the projected potential evaporation estimates. The total uncertainty partitioning is composed by the sum of all specific variances of each modelling chain step (namely RCPs, GCMs, RCMs and the PE formulations), and a residual term, representing the interaction between models. Moreover, a 30-year rolling mean is applied on projected variables that are available from 1976 to 2099, to reduce the impact of internal variability. Within the QE-ANOVA analysis, the trend signal analysed by a variance decomposition is a trend model fitted to rolling mean projections. This statistical analysis allows to assess the importance of the choice in PE formulation, according to the time scale and geographical-area targets. Finally, a signal to noise ratio is used to determine the strengh of PE changes over France. The ensemble mean PE expected changes represents the signal, and the noise is represented by the standard deviation between simulations.

## 3 Results

### 3.1 Trends in potential evaporation according to the different RCPs

Figure 2a shows the mean annual PE, from multi-scenario and multi-model average, over the reference period 1976-2005 based on climate projections data. Across France, a north-south gradient is visible, with the highest values obtained over the Mediterranean region and the lowest ones over the northern part and the mountainous areas (Alps and Pyrenees).

Annual PE is shown to increase over the 1976-2005 period (Figure 2b). A mean increase from about 730 $mm.y^{-1}$ at the beginning of the century to about 840 $mm.y^{-1}$ by the end of the century is simulated for all RCPs averaged (Figure 2b). However, this PE increase highly depends on RCPs, as PE reaches around 775, 820, and 920 $mm.y^{-1}$ for RCPs 2.6, 4.5 and 8.5, respectively. This means that for the majority of climate models and PE formulations, the projected increase in greenhouse gas concentrations clearly leads to an increase in PE estimates and the trend in PE is closely related to the RCP considered: an exponential increase for RCP 8.5, a rather linear increase for RCP 4.5 and a plateau reached by 2040 for RCP 2.6.

### 3.2 Behavioural differences between PE formulations and links to climate variables

The PE formulations show large differences in terms of magnitude, whatever the time period considered (Figure 3a). This is in line with previous studies that compared PE amounts depending on the PE formulation (Federer et al., 1996; Kingston et al., 2009). The differences between formulations reach about 400 $mm.y^{-1}$, which is much higher than the expected PE changes over the 1976-2099 period for a given formulation.

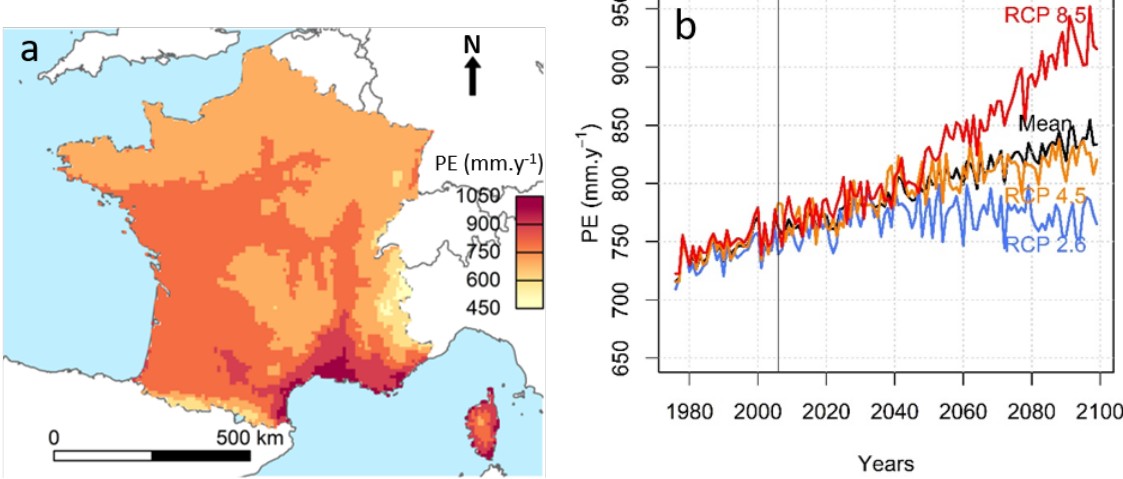

**Figure 2.** Mean annual PE ($mm.y^{-1}$) computed with climate projections from 1976 to 2005 plotted over France (a), and mean annual time series of PE averaged over the whole study area (b) computed with the three RCPs (1976-2099). Pluriannual (a) and annual (b) mean PEs were calculated by averaging the seven PE formulations and also the different GCM/RCM couples. As mentioned in Table 1, the RCP 2.6 time series is obtained by averaging 8*7 time series (8 GCM/RCM couples and 7 PE formulations), the RCP 4.5 time series is obtained by averaging 10*7 time series (10 GCM/RCM couples and 7 PE formulations) and the RCP 8.5 time series is obtained by averaging 12*7 time series (12 GCM/RCM couples and 7 PE formulations).

Figure 3b shows gradual positive anomalies for each formulation. By the end of the century, the annual PE changes over the 1976-2099 period are between +30 $mm.y^{-1}$ (in average over 30-year periods) with the Morton formulation and +130 $mm.y^{-1}$ with the Hamon formulation, whose relatively high sensitivity to air temperature increase was already demonstrated on other locations (Duan et al., 2017). However, interestingly, the growth rate does not necessarily depend neither on the selected forcing
variables nor on the type of equation, since Penman, Penman-Monteith and Oudin present similar trend slopes, while probably being the most different in terms of formulation and on the selected forcing variables.

To better understand this point, we analysed the link between the PE and climate variables anomalies, regardless of their use in PE formulations. We show that whatever the PE formulation, an increase in air temperature or radiation or a decrease in relative humidity leads to an increase in PE (Figure 4). The relationships exhibit rather linear behaviors, with higher slopes
obtained for the Hamon PE formulation. This suggests that the covariance between climate variables from GCM/RCM couples of models allows for PE formulations not making use of all available climate variables to still show consistent evolutions of annual PE. Surprisingly, no clear relationship is observed between the PE and wind speed anomalies. A large discrepancy exists between the GCMs/RCMs climate models regarding the sign of the evolution of wind speed, suggesting large uncertainties for this variable.

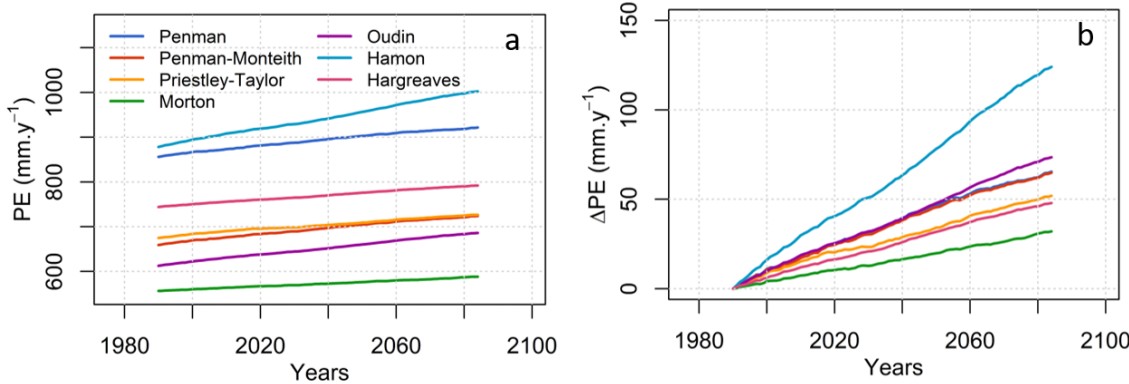

**Figure 3.** Projected mean annual PE (a) and expected increase (b) for each formulation. The time series are obtained by averaging the 30 RCP/GCM/RCM time series for each PE formulation. A 30-year rolling mean is applied so that, for instance, the value for the first year in the plotted time series (at year 1990) corresponds to the PE averaged over the 1976-2005 period.

## 3.3 Quantifying and partitioning the total uncertainty in PE projections

We showed in the previous section that differences between PE evolutions are large in the future. In this section, we aim at deciphering the real contribution of PE formulations to the total uncertainty, relatively to other uncertainty contributors (RCPs, GCMs and RCMs). Figure 5 shows the contribution of each modelling step (RCPs, GCMs, RCMs and PE formulations) to the total uncertainty of the future PE changes, i.e. the proportion of the variance explained by each step in the modelling chain at three different 30-year periods centered around 2030, 2050 and 2085, over the entire domain of the study (France), obtained applying the QUALYPSO method.

By 2030, GCMs are the main source of uncertainty with a contribution ranging from 30% (over the south-east part of the domain) to 50% (over the north-west part of the domain). RCMs are the second largest source of uncertainty with a contribution ranging from 20% to 30%. The other steps in the modelling chain exhibit a much lower contribution: PE formulations account for less than 5% to the total uncertainty, except over the south-east part of the domain, where it reaches 10%. The RCPs represent less than 10% of the total uncertainty. Finally, the residual is quite low with a proportion below 10%.

By 2050, GCMs are still the largest source of uncertainty over a large part of the territory, with a contribution of about 40% to the total uncertainty. However, in the southern part of the territory, this contribution is still lower (30%). The lower contribution of GCMs in this part of the territory can be explained by the increase in the contribution of RCPs to the total uncertainty, on average by up to 10%. The contributions of RCMs show a small increase in the center of the French territory, from 30% to 40% on average. The contributions of other steps remain similar as those for 2030.

By 2085, GCMs and RCMs provide the major sources of uncertainty in the modelling chain, with a contribution of 30% each. The contribution of RCPs to the total uncertainty is particularly higher than for other periods (especially in the south part of the domain), so that in 2085, the divergence between RCP scenarios explains 20% of the variances in PE estimates. Finally,

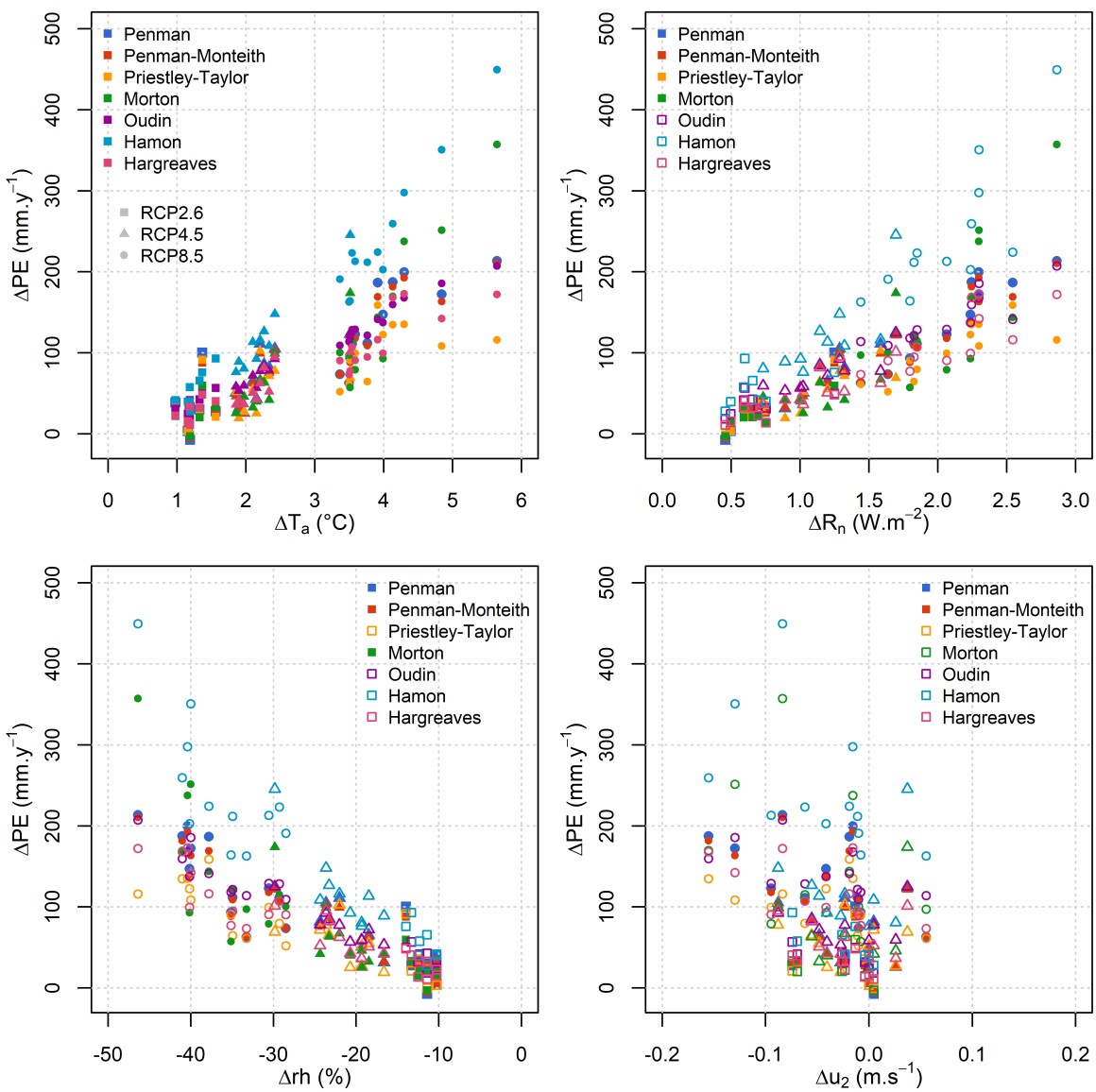

**Figure 4.** Mean anomalies of annual PE compared to the average anomalies of the climate variables by the end of the century (2070-2099), for air temperature ($T_a$, top left), net radiation ($R_n$, top right), relative humidity ($rh$, bottom left) and wind speed ($u_2$, bottom right). Each symbol represents one combination of RCP/GCM/RCM/PE formulation. Full symbols are used when the climate variable is used in the PE formulation, blank symbols are used when the climate variable is not used in the PE formulation.

differences between PE formulations, although they lead to largely different PE changes, remain a minor source of uncertainty compared to other factors, with a contribution to the total uncertainty below 10%.

The total variance of PE change is the total uncertainty of PE response to climate change. Interestingly, the total uncertainty increases with time, especially for the southern regions, where the contribution of RCPs is relatively high.

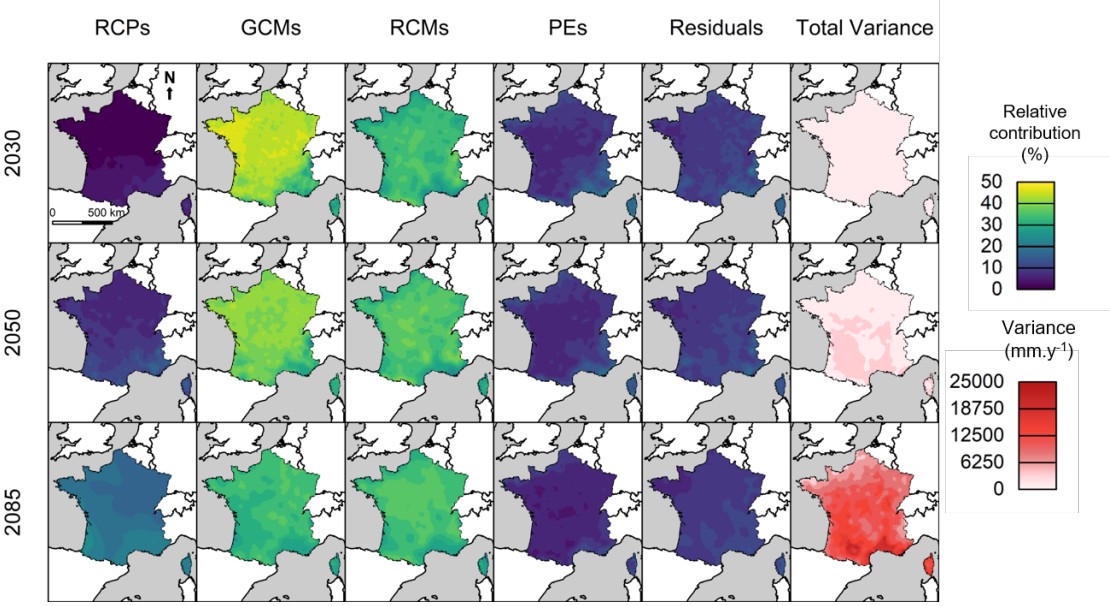

**Figure 5.** Uncertainty contribution of each impact modelling chain step, to changes in PE across France at three different 30-year periods centered around 2030, 2050 and 2085. Contribution to the total uncertainty is presented in percents and is calculated with the QUALYPSO method.

## 3.4 Analyzing the uncertainty in PE projections from a single scenario: the RCP 8.5

The contribution of PE formulations to the PE projection uncertainty might be underestimated by the combining of RCPs in the previous analysis. As a consequence, the contribution of PE formulations to the PE projection uncertainty is assessed here for RCP 8.5 only. Figure 6 presents the results of the uncertainty analysis for a single scenario: RCP 8.5, by 2085. According to the results presented in figure 5, the contribution of RCPs becomes higher for the end of the century, compared to earlier periods, i.e. when RCPs convey more diverging climate signals regarding RCP 8.5, the total variance (Figure 6) is moderately

smaller than in the multi-scenario analysis (Figure 5). Considering multiple RCPs in the long term therefore introduces more variability. Figure 6 shows that the relative contribution of the PE formulations is higher for the analysis considering only RCP 8.5 than for the multi-scenario analysis. This is especially the case for the southern locations where the PE contribution reaches almost 50% of the variance. GCMs and RCMs contributions remain stable between the single and multiple scenario analysis.

The results of the uncertainty analysis for a single scenario highlight the important contribution of the PE formulations over the total uncertainty (RCPs excluded), and even rank the PE formulations as the first uncertainty contributor for south of France.

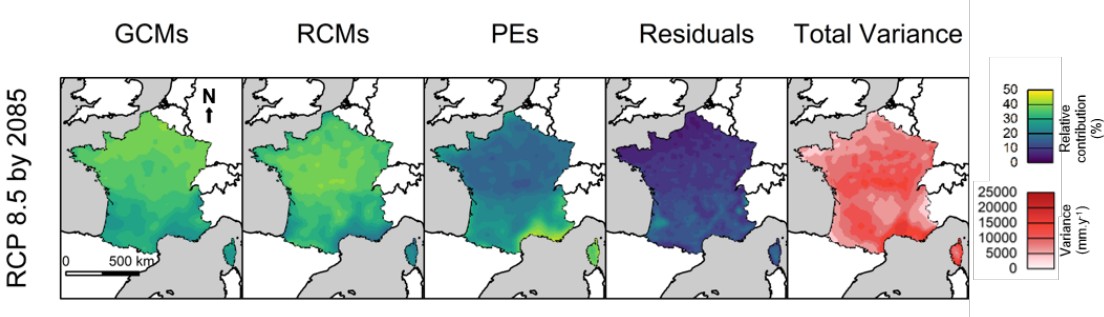

**Figure 6.** Uncertainty contribution of each impact modelling chain step to changes in PE across France at a 30-year period centered around 2085, for the RCP 8.5. Contribution to the total uncertainty is presented in percents and is calculated with the QUALYPSO method.

Figure 7 (top row) shows the mean yearly PE increase ($mm.y^{-1}$) for RCP 8.5 computed with the complete matrix of data, i.e. completed with the QUALYPSO framework, at three different 30-year periods centered around 2030, 2050 and 2085. The PE increase for future lead times, computed from the inferred complete matrix with the augmentation process, does not modify the spatial distribution of the original data as shown in Figure 2. The south-north gradient is still well represented for PE anomalies, only mountainous areas are no longer standing out.

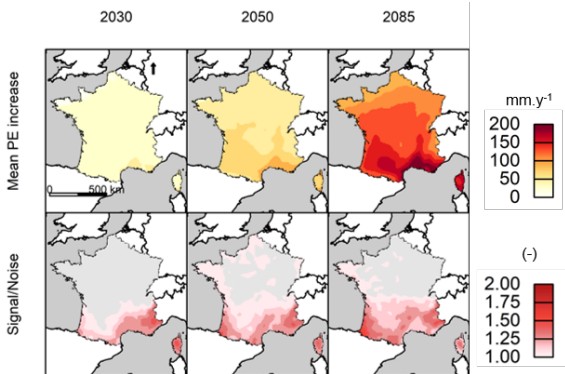

**Figure 7.** Mean change in annual PE ($mm.y^{-1}$), computed with the QUALYPSO framework (top row); signal-to-noise ratio (bottom row). Results are shown at three different 30-year periods centered around 2030, 2050 and 2085. Grey areas indicate a signal to noise ratio below 1 at the 90% confidence bounds, and vice versa for red areas.

Finally, the signal-to-noise ratio due to climate change, based on the method of Hawkins and Sutton (2012), is presented in Figure 7 (bottom row). This indicator allows assessing if the change signal is more important than the uncertainty associated to the change trend (i.e. the noise of the ensemble). When the signal-to-noise ratio is above one, the change is more important than

the uncertainty and an emergence time can be associated to the change trend. The expected increase in PE becomes greater than
the variability for the southern areas of the domain, indicating a signal emergence. For northern locations a large uncertainty is
associated to the estimation of future PE.

## 4    Discussion

### 4.1    Investigating the role of the uncertainty partitioning approach on results

We performed a descriptive analysis to evaluate the distribution of PE changes, more specifically the distributions of PE
anomalies averaged over the entire domain (France) according to each scenario and model. Figure 8 shows the variability of
the mean PE anomaly projected over France for each model and scenario used in the study, namely RCPs, GCMs, RCMs and
PE formulations, at three different 30-year periods centered around 2030, 2050 and 2085. The figure allows identifying which
scenario or model in the ensemble is likely to convey more uncertainty to the projections. For example, a model showing a
greater variability than the other ones or a significantly different projection can be identified as contributing to a significant
amount to the ensemble uncertainty range. At a first glance, the figure shows the increase in variability of the projections with
time, i.e. a growing uncertainty associated to the PE projections from horizon 2030 to horizon 2085. This increase is relevant
to previously shown results, such as the divergence between RCPs (Figure 2), the divergence between PE formulations (Figure
3) or the increase in total uncertainty (Figure 5).

Regarding the RCPs, the divergence between the three of them increases with time. At the 2085 horizon, PE projections
clearly differ according to the emission scenario, which explains results observed in Figure 5 (increased proportion of uncer-
tainty due to RCPs). As expected, RCP 8.5 shows the largest uncertainty range and mean value increase, followed by RCP 4.5
and RCP 2.6. Given that the signal-to-noise ratio (in Figure 7) is calculated from the mean of the total set of projections, we
can assume that if RCP 8.5 only were considered to estimate future PE, the signal-to-noise ratio would be greater and be more
likely to become above unity on some areas and under some horizons (as shown in Figure 6), while considering RCP 2.6 only,
the signal-to-noise ratio would be lower.

Regarding the GCMs, the divergence of PE projections between each model increases over time in terms of median change
and distributions of change. Moreover, the uncertainty spread also increases with time, which adds up to the total uncertainty,
despite the fact that the relative contribution of this factor to the total uncertainty decreases with time (see Fig. 5). Two
GCMs stand out from the rest: ICHEC-EC-EARTH projects particularly higher PE than the others, while MPI-ESM-LR shows
significantly lower PE values.

The divergence between RCMs increases over time as well. It must be highlighted that not all RCMs are used with the
same GCMs. For example, CLMcom-CCLM4-8-17 appears in Figure 8 as the RCM with the most variability. However, this
RCM only uses the outputs of two different GCMs, the MPI-ESM-LR model that projects the lowest PE increase, and the
MOHC-HadGEM2-ES model that projects one of the highest PE increases. The projected higher uncertainty derived from this
RCM can thus be explained by significantly divergent GCM inputs to this model compared to the other RCMs.

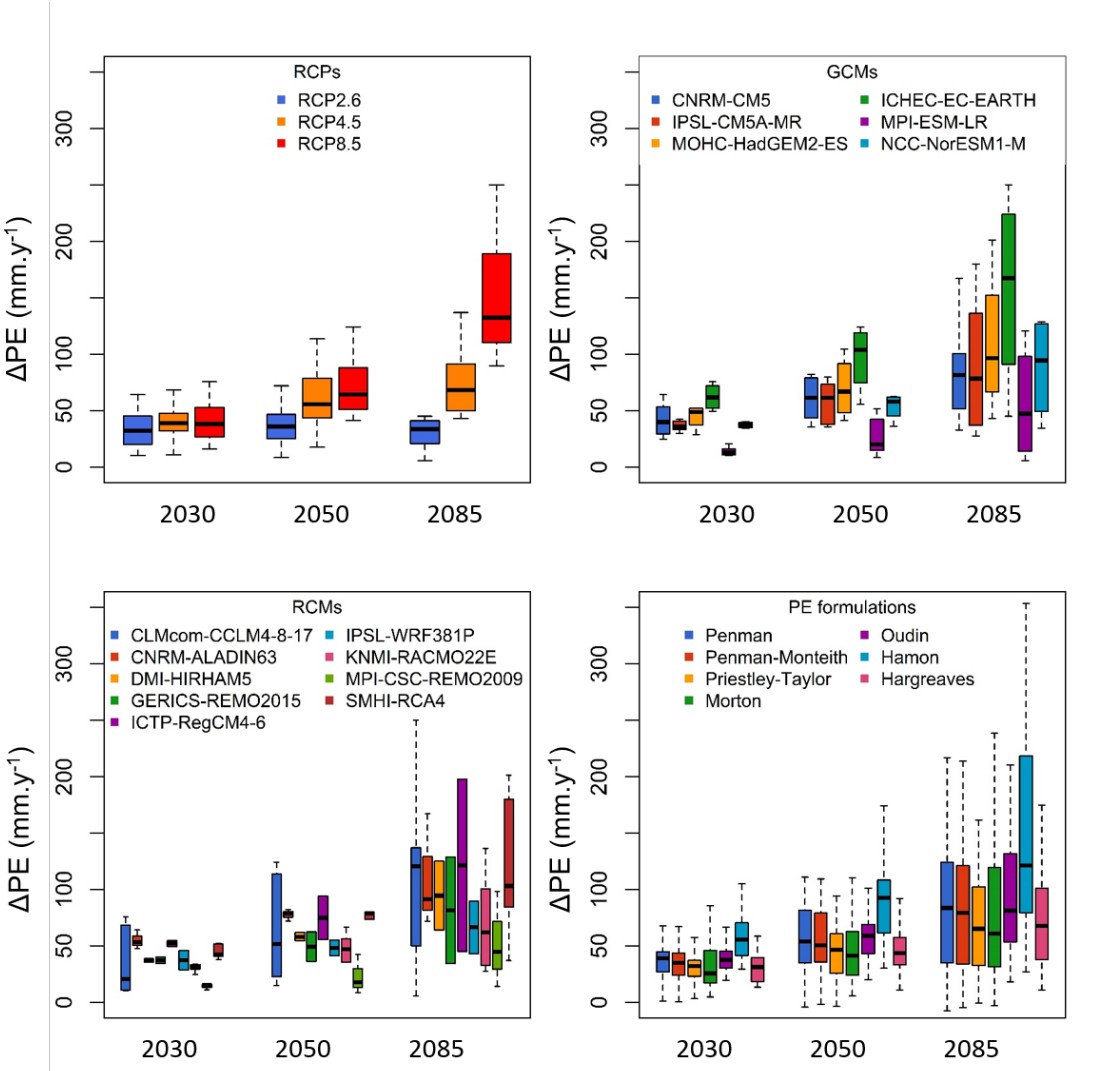

**Figure 8.** Distribution of annual PE changes across France at three different 30-year periods centered around 2030, 2050 and 2085 for each modelling step: RCPs (top left), GCMs (top right), RCMs (bottom left) and PE formulations (bottom right). The boxplots represent the quantiles 5, 25, 50, 75 and 95 of the distributions. A boxplot gives informations on the variability in PE anomaly, for a given modelling step, while the other modelling steps vary across their PE anomalies range.

Finally, Figure 8 also shows changes in PE projection uncertainty level and spread derived from the PE formulations. For each time horizon, the uncertainty level and spread show similar values across the seven PE models. However, for each formula, the variability and hence the uncertainty to PE projections increases over time. In addition, it is clear that except the Hamon formulation, all PE projection anomalies remain within similar value ranges. This important result confirms that PE formulations are not the main factor of uncertainty in PE projections, comparatively to RCPs, GCMs and RCMs.

## 4.2 Guidelines for selecting PE formulations in impact studies

Two related questions arise when using PE as inputs of climate change impact models: (1) do we need to consider multiple PE formulations or does a single PE formulation suffice? and (2) which formulation(s) are preferable?

As for the first question, our results suggest that PE formulations may have different future trends and assessing these differences through a multi-formulation approach is undoubtedly a relevant practice. The PE anomalies obtained in this study are of the same order of magnitude than those of previous studies and rank PE formulations in a similar way (Milly and Dunne, 2016, 2017). However, uncertainty analysis results depend on the framework of the climate impact study: combining multiple scenarios or treating each scenario independently. For the multi-scenario analysis, considering the other sources of uncertainties conveyed by RCPs, GCMs and RCMs, we found that PE formulations account for only around 10% of the total uncertainty in PE projections. This result contrasts with previous studies that highlighted the relatively important role of PE formulation in the impact modelling chain (Vidal et al., 2013; Seiller and Anctil, 2016; Williamson et al., 2016). However, we must notice that many studies focused on the RCP 8.5 scenario, which is associated to the highest greenhouse gas emissions and therefore the highest radiative forcing. The results from the uncertainty analysis performed for the RCP 8.5 are more consistent with other studies. This leads to greater divergence between the formulations than in the multi-scenario approach, due to the higher future air temperature gradient. We attribute the differences between our results and those from previous studies to the inclusion of several climate model simulations under a large range of emission scenarios and to the approach used to partition uncertainty relying on a Bayesian method to complete our unbalanced set of projections. However, another hypothesis that could explain the discrepancy between our results and other studies is the use of bias-corrected data or not. Using bias-corrected data reduces the contribution of GCMs/RCMs to total uncertainty (since their distributions are forced to match an observed data set), in contrast to using raw data. In this study, we prefer not to include bias-corrected data, because most bias-corrected climate projections with statistical methods perform poorly in retaining the properties of the covarying variables needed for the PE calculation (Vrac and Friederichs, 2015). In addition, oppositely to these studies, we did not use any impact model (such as a hydrological model or a crop water requirements model), aiming at making this study as general as possible. Due to the minor contribution of PE formulations to the total uncertainty in PE projections, when multiple scenarios are considered, it does not seem necessary to consider several formulations in multi-scenario climate impact studies. However, for a single scenario analysis (treating each scenario independently) the contribution of PE formulation becomes higher, and it seems necessary to quantify this source of uncertainty as well as the GCM and RCM contributions. And, in case several impact models are used, due to the differences in the magnitude of PE and its changes between formulations, the same formulation should be conserved for all modelling chains. These recommendations do not apply in case only one climate projection is used, which should be avoided anyway.

As for the second question, it is difficult to provide clear guidelines. In theory, the choice of a PE formulation for an impact study needs to consider both the uncertainties on the projected climate variables used by the formulations and the physical consistency handled by these formulations. In practice, since the choice of a PE formulation is often driven by the own experience of the modeller, the relatively low contribution of PE formulations, in multiple scenarios analysis, relatively to the

other sources of uncertainty, tends to confirm this practice. We showed that the behaviors of the PE formulations along projected climatic gradients are generally consistent (Fig. 4), even for very different formulations in terms of climate variables used and physical consistency. This results from the fact that other climate variables co-vary with temperature and these relationships are maintained in climate projections. If one wants to take into account other aspects such as $CO_2$, then we would recommend choosing a Penman-Monteith type formula that explicitly allows for this. We found that Penman-Monteith PE trends are near of the average of the PE formulations tested. Besides, its physical soundness allows taking into account interactions between climate drivers more explicitly than other formulations (even though we showed that those interactions tend to be implicitly taken into account in all formulations). In addition, its structure theoretically allows accounting for other environmental changes such as land use or plant behavior under elevated atmospheric $CO_2$ in a more explicit way (Schwingshackl et al., 2019; Yang et al., 2019). Accounting for these changes probably represents a greater challenge than identifying the "best" PE formulation with fixed vegetation parameters.

### 4.3  Implications of evaporation uncertainties in impact studies

This paper has tackled the issue of uncertainty in PE projections, but PE is only a modelling results. We discuss here how PE uncertainty is transfered to actual evaporation (AE) uncertainty. Generally, AE is computed as a fraction of PE, typically constrained by soil moisture. How uncertainty in PE propagates to AE depends on regions. In water-limited regions (such as in the Mediterranean region in our study), the impact of PE uncertainty on AE estimate is negligible. However, the precipitation projections have a large uncertainty, which affects the estimated AE in such regions. Indeed, an increasing (resp. decreasing) trend of precipitation results in increased (resp. decreased) soil moisture, and thus AE. In energy-limited regions, the uncertainty of PE is more important than the uncertainty of precipitation to estimate long-term AE.

This leads to questioning the sensitivity of impact models to PE variability. Numerous rainfall-runoff models used for impact studies use PE as input. The representation of the evaporation process in the model may have consequences for the transfer of uncertainties. For example, HBV model considers AE equal to PE if soil moisture is enough, if not AE is reduced to satisfy the previous condition, which could reduce PE uncertainty (Seibert and Vis, 2012). Calibrated hydrological models can accommodate errors in PE estimates (Oudin et al., 2006) but are much more sensitive if the errors are not constant over time (Nandakumar and Mein, 1997). The optimisation of any hydrological model parameters model with observed streamflow data might compensate errors and uncertainty on PE. This becomes an issue for the extrapolation period, where the risk is to project excessive dry or wet future conditions under climate change. While the equations described in this paper are used for rainfall-runoff models, and thus for hydrological climate impact studies, they are not necessarily appropriate for crop water requirements as crop specific parameters might be considered. For such a purpose, the set of PE formulations should represent vegetation parameters, which could lead to different evolutions than for the set of formulations chosen in this study. The crop specific parameters are indeed susceptible to be sensitive to other aspects than only the climatic variables tested in our study, which might modify the uncertainty contribution.

## 5  Conclusions

Potential evaporation (PE) is a necessary proxy for the estimation of actual evaporation in impact models such as hydrological and crop water requirements models. However, many formulations exist, and the role of these formulations on projections remains largely unclear. We investigated in this study the uncertainties sources of PE projections using 30 RCP/GCM/RCM combinations and seven PE formulations over the 21$^{st}$ century. Thanks to an ANOVA-based method, we assessed the contribution of each step of the PE modelling chain to PE uncertainty.

Our work shows that whatever the PE formulation used, the mean PE will increase across France (from +40 to 130 $mm.y^{-1}$ in average over 30-year periods), with higher increases associated to a higher greenhouse gas emission scenario. The PE increase is higher on the southern part of France and lower on the northern part. Moreover, uncertainties are large, leading to a signal-to-noise ratio higher than one for southern locations, and below one for northern locations. This involves issues to determine an emergence time in terms of signal change for the entire domain. The contribution of the PE formulations to the overall uncertainty in PE projections is much lower than the contribution of the other uncertainty sources (RCPs, GCMs and RCMs, namely) in a multiple scenario approach. On the other hand, in a single scenario analysis, with RCP 8.5, the contribution of the evaporation formulations becomes higher and even dominant for the southern part of France. However, this work also highlighted differences both in terms of absolute values and future changes in PE among the different formulations. The divergence between formulations was found to be higher with higher air temperature increases. Hamon formulation leads to the highest increase while Morton formulation leads to the smallest increase.

Finally, the ranking of formulations (according to magnitude) does not change over time, and the quantification of PE uncertainty contribution should depend of the framework single scenario or multiple scenarios. However, the bias induced by the formulation could lead to a different estimation depending on the impact modelling chain used. Further work may consider testing these conclusions with a full climate impact modelling chain, for example including an integrated hydrological model, which would have a different sensitivity to PE.

*Code availability.*  The PE formulation codes can be retrieved from: https://doi.org/10.15454/NCNCHG

*Data availability.*  EURO-CORDEX projections can be retrieved from https://www.euro-cordex.net/.

*Author contributions.*  All authors conceived the experimental set up. Thibault Lemaitre-Basset performed the calculation and the analysis and wrote a first version of the manuscript. All authors contributed to the final version of the manuscript.

*Competing interests.*  The authors declare no competing interests.

*Acknowledgements.* The authors acknowledge Météo-France for preparing the EURO-CORDEX climate projections on the right map pro-
jection. The first author was funded by Sorbonne University and by Agence de l'Eau Rhin-Meuse.

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
