# Peer review of "Unravelling the contribution of potential evaporation formulation to uncertainty under climate change"

_Hydrology and Earth System Sciences, 2021_

## Author Response (AR1)

**List of relevant changes made in the manuscript**

We include all the specific comments mentioned by the referee, clarify the vocabulary, and correct figures directly in the manuscript. We detail major changes made in the manuscript following referees suggestions, see the marked-up version of the revised manuscript.

We add a specific figure (Fig. 1) to clarify the method section. The figure represents the complete modelling chain used in this study with each modelling step (RCPs, GCMs, RCMs and PE formulations), and the uncertainty method analysis used.

The Table.2 was modified accordingly to the comments of referees; we provide more details about PE formulations with a new column 'type of application", and specific comments about Penman-Monteith FAO and Hamon formulation.

We move details into material and method part as recommended by referees about: the data set descriptions and the signal-to-noise method.

For the uncertainty analysis, we add a panel on figures with the total uncertainty, as suggested by the referees.

We follow the reviewer's recommendations; we analyse the new partition of the total variance for each factor (GCM/RCM/PE formulations) through conducting one uncertainty analysis on a single RCP, namely RCP 8.5. It provides new insights on the signal/noise ratio and its interpretation. RCP 8.5 has the strongest change signal of the three RCPs, and all the GCM/RCM couples used are available for this scenario. The results of this experiment are presented in a new sub-section "3.4: Analyzing the uncertainty in PE projections from a single scenario: the RCP 8.5". The discussion part is also modified to confront results from multi-scenarios and single scenarios approaches.

As proposed by referees we add a new discussion section dedicated to some qualitative statements on the transfer of PE uncertainty to AE, and the sensitivity of the impact model to PE uncertainty, namely "4.3 Implications of evaporation uncertainties in impact studies".

---

## Author Response (AR2)

We would like to thank again the referee and the editor for their useful comments and their interest for our study. We included in the manuscript all the specific comments mentioned by both of them.

REFEREE 1

Comment 1 :

Dear authors,
I think the changes and corrections improve the quality of your manuscript. Thank you for your efforts in implementing them. The new chapter 4.3 does not fulfill my expectations. I make the following comments/recommendations:

L302 : relatively low

Answer to comment 1:

We modified the manuscript as suggested.

Comment 2:

L318 : ..., typically constrained by soil moisture.

Answer to comment 2:

We modified the manuscript as suggested.

Comment 3:

L320 - 322 : I think these statements could be explained better. In Mediterranean regions, where PE is much higher than P (resulting in water-limited ET), AE uncertainty generally will not be affected by PE uncertainty but by P uncertainty. (The authors mention P uncertainty but I feel the connection to AE uncertainty is not explained properly). In energy-limited regions, PE uncertainty might be transferable to modelled AE uncertainty 1-to-1.

Answer to comment 3:

We improved the explanations regarding this point, and we proposed the following reformulation of sentences from "The role of PE…" to "…among different climate models":

"How uncertainty in PE propagates to AE depends on regions. In water-limited regions (such as in the Mediterranean region in our study), the impact of PE uncertainty on AE estimate is negligible. However, the precipitation projections have a large uncertainty, which affects the estimated AE in such regions. Indeed, an increasing (resp. decreasing) trend of precipitation results in increased (resp. decreased) soil moisture, and thus AE. In energy-limited regions, the uncertainty of PE is more important than the uncertainty of precipitation to estimate long-term AE. "

Comment 4:

L326 : delete "this"

Answer to comment 4:

We modified as suggested.

Comment 5:

L331 - 334 : I am not sure that what is hypothesized on is valid. Most PE equations are actually designed for crop ET, applying them on catchment scale or for AE estimation in rainfall-runoff models actually reduces the validity of these equations.. I would say that applying these equation for hydrological models at catchment scale is as "inappropriate" as not considering crop specific parameters for crop water requirements. It is not clear to me what the authors want to say in this paragraph.

To me, AE uncertainty will depend on many more factor than PE uncertainty. Maybe this is the point the authors are trying to make..

We want to explain that these conclusions are drawn for PE formulations that didn't account for crop parameters

Answer to comment 5:

Section 4.3 explains the implications of our conclusions on the uncertainty of evapotranspiration formulas. Several impact studies (which care about evaporation and water resources) are interested in the agronomic consequences of climate change (irrigation needs, land use changes …). These studies use equations with parameters to take into account the vegetation. For this reason, we found appropriate to remind that our conclusions did not take into account the type of vegetation, and that we had only explored the uncertainty of the PE formulations with the changes of the climatic variables.

We propose the following reformulation:

"While the equations described in this paper are used for rainfall-runoff models, and thus for hydrological climate impact studies, they are not necessarily appropriate for crop water requirements as crop specific parameters might be considered. For such a purpose, the set of PE formulations should represent vegetation parameters, which could lead to different evolutions than for the set of formulations chosen in this study. The crop specific parameters are indeed susceptible to be sensitive to other aspects than only the climatic variables tested in our study, which might modify the uncertainty contribution."

Editor's comment:

In several figures (e.g. Fig.8), but also in the text (e.g. Conclusion section) the change of PE, i.e. delta_PE is given in units of mm/yr. I may have misunderstood something here, but a delta_PE of ~100 mm/yr would entail a total increase until the end of the century of PE of ~10000 mm/100years (!!!). I suppose this is not what we are seeing here. Please correct the units or provide a clear explanation of the meaning of that unit

Answer to editor's comment:

We proposed a clearer explanation on the meaning of that unit in the text. Actually, we display annual PE, which is therefore expressed in mm.y-1. The anomaly (or change between two periods) is therefore also expressed in mm.y-1. We agree that an increase of 100 mm every year is not realistic, but this is not what we meant: we meant that the change in annual PE over an average year is around 100 millimeters, i.e. 100 mm.y-1.

We hope that the additional explanations in the figure captions and in the text make that clearer.